# Unsupervised Model Tree Heritage Recovery

**Eliahu Horwitz, Asaf Shul, Yedid Hoshen**
School of Computer Science and Engineering
The Hebrew University of Jerusalem, Israel
{eliahu.horwitz, asaf.shul, yedid.hoshen}@mail.huji.ac.il
https://horwitz.ai/mother

## Abstract

The number of models shared online has recently skyrocketed, with over one million public models available on Hugging Face. Sharing models allows other users to build on existing models, using them as initialization for fine-tuning, improving accuracy, and saving compute and energy. However, it also raises important intellectual property issues, as fine-tuning may violate the license terms of the original model or that of its training data. A *Model Tree*, i.e., a tree data structure rooted at a foundation model and having directed edges between a parent model and other models directly fine-tuned from it (children), would settle such disputes by making the model heritage explicit. Unfortunately, current models are not well documented, with most model metadata (e.g., "model cards") not providing accurate information about heritage. In this paper, we introduce the task of *Unsupervised Model Tree Heritage Recovery* (Unsupervised MoTHer Recovery) for collections of neural networks. For each pair of models, this task requires: i) determining if they are directly related, and ii) establishing the direction of the relationship. Our hypothesis is that model weights encode this information, the challenge is to decode the underlying tree structure given the weights. We discover several properties of model weights that allow us to perform this task. By using these properties, we formulate the MoTHer Recovery task as finding a directed minimal spanning tree. In extensive experiments we demonstrate that our method successfully reconstructs complex Model Trees.

## 1 Introduction

The number and diversity of neural models shared online have been growing at an unprecedented rate. For instance, on the popular model repository *Hugging Face* alone there are over one million models, with thousands more added daily. Many of these models are related through a common ancestor (i.e., foundation model) from which they were fine-tuned. Recovering the heritage of a model is important for model and data attribution. Determining whether one model originated from another (i.e., via fine-tuning) can help resolve legal disputes over model authorship in cases where each party claims the other model is based on their proprietary model. Moreover, it can help identify models that resulted from the wrongful use of proprietary training data.

Public models provide 3 main sources of information: metadata, architecture, and weights. While model metadata may specify model heritage explicitly, the majority of public models lack proper documentation (e.g., empty model cards). Concretely, we found that over $60\%$ of models on *Hugging Face* are missing information about their heritage (see App. A for a detailed analysis). Also, as many unrelated foundation models share the same architecture (e.g., ViT-B (Dosovitskiy et al., 2020)), we cannot predict heritage by looking at the model architecture. As weights of public models are very expressive and always available, in this paper we use just the weights.

Motivated by Darwin's tree of life (Darwin, 1859), which describes the relationships between organisms, we analogously aim to discover the *Model Tree*, a structure that describes the hereditary relations between models. Reminiscent of the natural world, the structure of the Model Tree is unknown. We therefore propose the task of *Unsupervised Model Tree Heritage Recovery* (Unsupervised MoTHer Recovery) for mapping Model Trees in the rapidly evolving neural network ecosystem.

We begin our exploration of the Model Tree by studying the relationship between the weights of related models. First, we establish that the distance between the weights of a pair of models correlates with their node distance on the Model Tree. We then proceed to examine how the weights of a model evolve over the course of training. We observe that the number of weight outliers changes monotonically over the course of training. Specifically, we distinguish between a generalization stage (often called pre-training) and a specialization stage (often referred to as fine-tuning). We find that during generalization, the number of weight outliers grows, while during specialization, it decreases.

With these observations, we recover a Model Tree for a given set of models. This requires determining whether each pair of models is directly connected and establishing the direction of the relationship. We use weight distance to create a pairwise distance matrix between models and the outlier monotonicity to create a binary edge direction matrix. Finally, we use a minimum directed spanning tree algorithm on the combined distance matrix to recover our Model Tree. We extend the Model Tree to a Model Graph, allowing us to recover ecosystems with multiple Model Trees by first clustering the nodes based on their pairwise distances. To evaluate the MoTHer Recovery task, we introduce the MoTHer dataset, a Model Graph comprising of over $500$ models from diverse architectures and modalities.

To summarize, our main contributions are:

1. Introducing the task of Unsupervised Model Tree Heritage Recovery for model attribution.
2. Uncovering a connection between model weights and their relative location on Model Trees.
3. Proposing MoTHer, a new approach for unsupervised Model Tree Heritage Recovery and demonstrating its effectiveness.

## 2 RELATED WORKS

The field of weight space learning treats models as data points and attempts to extract information about a model by processing its weights. Eilertsen et al. (2020) performed a large empirical study of model weights, representing each model as a point in the *neural weight space* and training classifiers to predict different training hyperparameters. The seminal work of (Kong et al., 2016) proposed using a HyperNet to generate parameters of a model, recent works (Erkoç et al., 2023; Zhang et al., 2024; Wang et al., 2024; Peebles et al., 2022) used diffusion models to generate the weights of small networks. A different line of works predicts model performance or classifies models based on their weights by projecting the weight into an embedding space (Dar et al., 2022; Gueta et al., 2023), learning model weights representations (Schürholt et al., 2021; 2022; 2024), or by incorporating permutation invariance priors while training a neural network on the weights (Zhou et al., 2024; Kofinas et al., 2024; Lim et al., 2023; Navon et al., 2023b;a; Unterthiner et al., 2020). Recently, Horwitz et al. (2024) recovered the weights of a pre-trained model using a small number of LoRA fine-tuned models, which means that their safety is vulnerable. Most related to our work is Yu & Wang (2024) which uses model Jacobians to identify the parent model for a given model and a suspected set of models. However, it can not recover the direction of an edge or compute a complex heritage hierarchy. In addition, Yu & Wang (2024) uses the gradients of models, which are both very computationally expensive and require running samples (which are often unavailable) through the model. In contrast, our method is unsupervised, data-free, and can recover the structure of large and complex model collections. Overall, model weight processing remains an underexplored area that is poised to grow in the coming years.

## 3 MODEL TREES AND MODEL GRAPHS

This section describes the *Model Tree*, a data structure for describing the origin of models stemming from a base model (e.g., a foundation model) and defines the task of recovering its structure.

**Definition.** Consider a set of models $\mathcal{V}$, where the base model $v_b \in \mathcal{V}$ serves as the root node. Every model $v \in \mathcal{V} \setminus \{v_b\}$, is fine-tuned from another model. We refer to the model from which $v$ was fine-tuned as its *parent* model, and denote it by $Pa(v)$. Conversely, we refer to $v$ as a *child* of $Pa(v)$. A parent can have multiple children (including none), while all models except the root have only one parent. The set of tree edges is denoted by $\mathcal{E}$, where each directed edge between a parent and its child is represented as $e = (Pa(v), v)$. Overall, we define the Model Tree $\mathcal{T}$ by its nodes and

directed edges, $\mathcal{T} = (\mathcal{V}, \mathcal{E})$. We denote by $d(u, v)$ the number of edges on the shortest path in $\mathcal{T}_+$ between the nodes $u$ and $v$. The tree $\mathcal{T}_+$ is the same as our tree $\mathcal{T}$ except that the directed edges are replaced by undirected ones.

A collection of Model Trees $\mathcal{T}_1, \ldots, \mathcal{T}_n$ forms a forest, which we call a *Model Graph*. This Model Graph is defined as $\mathcal{G} = (\mathcal{V} = \mathcal{V}_1 \cup \ldots \cup \mathcal{V}_n, \mathcal{E} = \mathcal{E}_1 \cup \ldots \cup \mathcal{E}_n)$. In a Model Graph, $d(u, v)$ is only defined if $u, v \in \mathcal{T}_i$, when $u \in \mathcal{T}_i$ and $v \in \mathcal{T}_j$, $d(u, v)$ is undefined. Note that all the models within a Model Tree share the same architecture[1]. As the architecture of a model is given by its weights, and since different architectures necessarily belong to different trees, we can assume without loss of generality that all $v \in \mathcal{V}$ are of the same architecture.

**Task definition.** Due to the large number and diversity of models, the structure of the Model Graph is unknown and is non-trivial to estimate. We therefore introduce the task of *Model Tree Heritage Recovery* (MoTHer Recovery) for mapping the structure of the Model Graph.

Formally, given a set of models $\mathcal{V}$, the goal is to recover the structure of the Model Graph $\mathcal{G} = (\mathcal{V}, \mathcal{E})$ based solely on the weights of the models $v \in \mathcal{V}$. Since a Model Graph is a forest of Model Trees, the task involves two main steps: (i) Cluster the nodes into different components $\mathcal{T}_1, \mathcal{T}_2, \ldots$, where each component is a Model Tree with an unknown structure. (ii) Recover the structure of each Model Tree $\mathcal{T}_i$. Essentially, as each graph is defined by its vertices and edges, the task is to recover the directed edges $\mathcal{E}$ using the weights of $v \in \mathcal{V}$.

## 4 MODEL GRAPH PRIORS

Despite the recent growth in public models, and although model weights fully characterize the behavior of a model, our understanding of model weights is limited. Here, we explore two key properties of model weights that we use in Sec. 5 for Unsupervised Model Graph recovery (MoTHer).

### 4.1 ESTIMATING
NODE DISTANCE FROM MODEL WEIGHTS

In this section, we investigate the use of model weights to predict the distance between two models in the Model Tree. This will help us determine whether two models are related via an edge.

**Definition: weight distance between models.** Let $u$ and $v$ be two models, and $u_l$ and $v_l$ denote the weight matrix of layer $l$ of models $u$ and $v$ respectively,

$$\ell_{FT}(u, v) = \frac{1}{L} \sum_{l=1}^{L} \ell_2(u_l, v_l) \quad (1)$$

where $L$ is the number of model layers.

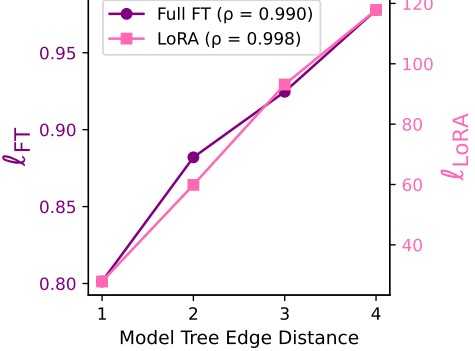

Figure 1: *Weight Distance vs. Model Tree Edge Distance:* For every pair of models, we plot the weight distance and the corresponding edge distance on the Model Tree. Our weight distances $\ell_{FT}$ and $\ell_{LoRA}$ almost perfectly correlate with the number of edges between models in a Model Tree. This correlation confirms these weight distances are good indicators for determining parent-child relation, i.e., models that were fine-tuned from one another. We use a 3 levels deep Model Tree that contains 21 models

**Full fine-tuning.** We first study the weight distance $\ell_{FT}(u, v)$ between pairs of models as a function of the edge distance $d(u, v)$ between their respective nodes on the Model Tree. In Fig. 1, we plot the relationship between these two distances ($\rho = 0.99$). It is evident that nodes with direct parent-child connections (i.e., models fine-tuned from one another) have the lowest weight distance of 1. We conclude that a low $\ell_{FT}$ distance between two models is highly correlated with an edge between their nodes and vice versa.

---

[1]While it is common to practice to add one or more layers to the end of a model and perform linear probing or fine-tuning, the pre-trained and fine-tuned models still share the "stump" architecture. In such cases, we can simply discard the disjoint layers and consider the intersecting layers.

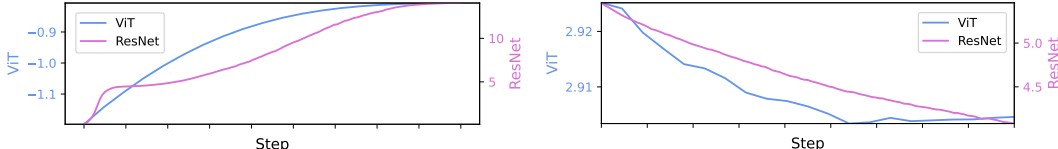

Figure 2: *Directional Weight Score:* We plot the change in the directional weight score throughout the pre-training (generalization) stage (left) and fine-tuning (specialization) stage (right). In all cases, the directional score is almost monotonic, indicating the increasing number of weight outlier values during generalization and the decreasing number during specialization. This confirms that our directional weight score is effective for determining the direction of an edge. For the fine-tuning, we used publicly available, pre-trained backbones as initialization

**LoRA fine-tuning.** LoRA (Hu et al., 2021) has become the dominant method for parameter-efficient fine-tuning. When fine-tuning a model via LoRA, the entire model is frozen, and only a subset of the layers tune a *new* low-rank matrix for each layer. Consequently, a model fine-tuned via rank $r$ LoRA differs from its base model by a matrix of at most rank $r$ for each layer. Furthermore, two models fine-tuned from the same base model using rank $r_1$ and $r_2$ LoRAs differ from each other by a matrix of at most rank $r_1 + r_2$ per layer. We use this property to provide a better estimate of the node distance between LoRA models and define the LoRA weight distance as:

$$\ell_{LoRA}(u, v) = \max_l \left( rank(u_l - v_l) \right) \tag{2}$$

where $L$ is the number of fine-tuned LoRA layers. In practice, we compute the rank using singular value decomposition (SVD), where the rank is the number of singular values greater than some threshold $\epsilon$. Similar to the full fine-tuning case (see Fig. 1), a low LoRA weight distance between two models is highly correlated with an edge between their nodes.

## 4.2 ESTIMATING EDGE DIRECTION FROM WEIGHTS

The direction of an edge between two nodes $u$ and $v$ reflects whether model $v$ was trained from $u$ or vice versa. The weight distance from Sec. 4.1 cannot disentangle the direction as it is symmetric. Estimating the direction of edges requires a statistic of the weights that evolve monotonically during training. To this end, we use kurtosis (i.e., fourth moment) and define the *directional weight score* as:

$$k(u) = \sum_{l \in L} \frac{E\left[ (u_l - \mu)^4 \right]}{\left( E\left[ (u_l - \mu)^2 \right] \right)^2} \tag{3}$$

where $L$ is a set of model layers and $\mu$ is the mean of the layer weights $l$. Note that the directional score only defines an order between related nodes; unrelated nodes may have very different scores.

To study the effectiveness of this score, we study how the weights of a model evolve throughout the training process. Concretely, we calculate Eq. 3 at multiple points throughout the training process and plot the results. Interestingly, we found that the training process can be categorized into two stages: a *generalization stage* and a *specialization stage*. As seen in Fig. 2, while the score is monotonic in both stages, it increases during generalization and decreases during specialization. The generalization stage usually corresponds with model pre-training and the specialization with model fine-tuning, however, this is not necessarily always the case. The term fine-tuning is typically used for any training performed after the initial pre-training stage. However, generalization training may also take place in an already pre-trained model (we show such an example in Sec. 6 (Stable Diffusion)). We therefore use the terms generalization and specialization.

**Intuition.** Consider a model with Gaussian-based weight initialization, such as Kaiming initialization by He et al. (2015). During the generalization stage, to better encode large, general-purpose datasets, the weights of a model will likely take an increasing number of diverse values. This may

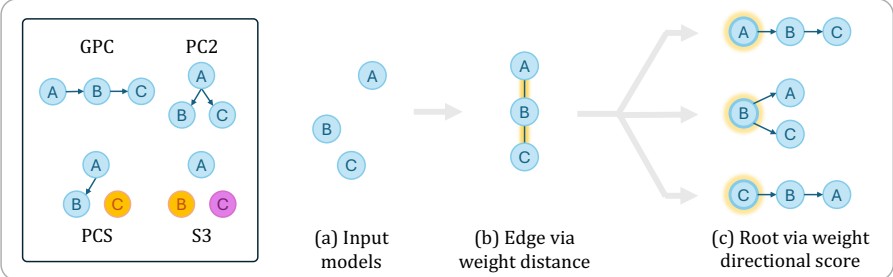

Figure 3: ***Recovering a Simplified Model Graph:*** We enumerate all possible Model Graphs of size 3 (left). On the right, we demonstrate a Model Graph Recovery process. (a) A set of 3 models with no prior knowledge regarding their relations. (b) Place edges between the nodes with the lowest weight distance. (c) Designate the node with the highest directional weight score as the root

lead to an increase in the number of outlier values in the weight matrix. In contrast, during the specialization stage, the weights of a model with a lower value diversity will likely suffice to encompass the smaller, task-specific data. This may lead to a decrease in the number of outlier values in the weight matrix. Kurtosis, as a measure of the "tailedness" of a distribution, captures this property.

## 5 MODEL TREE HERITAGE RECOVERY

Given the above priors, we now describe how to recover the structure of Model Graphs. As a warm-up problem, in Sec. 5.1 we start by manually solving all the 3 nodes Model Graphs. Then, in Sec. 5.2 we describe *MoTHer*, our proposed algorithm for recovering real Model Graphs.

In both cases, we have access to the model weights and assume that we are given the training stage of each node (i.e., generalization or specialization)[2]. Each Model Graph may contain one or more Model Trees. Connected nodes within a Model Tree are derived from each other via additional training steps. *Unless otherwise specified, we assume no prior knowledge regarding the model relations.*

### 5.1 WARM-UP: A SIMPLIFIED MODEL GRAPH

To recover a Model Graph of size 3, we place edges between the nodes with the lowest weight distance and designate the node with the highest directional weight score as the root. Next, we elaborate on each of the possible size 3 Model Graphs.

**Grandparent-Parent-Child (GPC).**  This Model Tree exhibits a 3-generational relationship, where each node is derived from the previous one (see Fig. 3). To recover the underlying Model Tree structure, we use $\ell_{FT}$ to place edges between node pairs with the lowest weight distances. This is motivated by our analysis in Sec. 4.1 and Fig. 1. Next, to determine the order of the nodes, we use the directional score shown in Eq. 3 and designate the node with the highest score as the root. Combining these steps fully constructs the GPC Model Tree, we illustrate this process in Fig. 3. Note that the above process assumes the specialization training stages for all 3 models. To adjust the process for the generalization stage, we can simply flip the sign of the directional score. When the training stages of each node differ, we choose the sign according to each node's training stage.

**Parent-Child-Child (PC2).**  This Model Tree contains one parent with two children (see Fig. 3). As in the GPC case, we start by placing the edges using the weight distance defined in Eq. 1. Since both children are derived from the root, the directional score will predict the node with the highest score as the root (see Fig. 3). Different training stages are handled as in the GPC case.

**Parent-Child-Stranger (PCS).**  Unlike the previous triplets, here we have a Model Graph comprised of two Model Trees (see Fig. 3). To recover this structure, we first identify the node with no edge

---

[2]This is a reasonable assumption as we generally know whether a model is a foundation model with general capabilities or a fine-tuned specialized model.

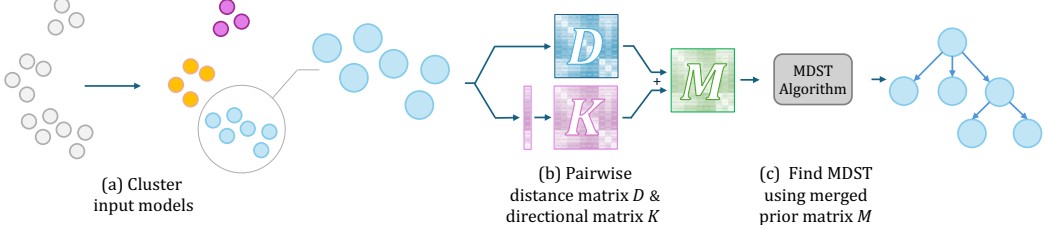

Figure 4: ***MoTHer Recovery Overview:*** Our proposed *Model Graphs* and *Model Trees* are new data structures for describing the heredity training relations between models. In these structures, heredity relations are represented as directed edges. We introduce the task of *MoTHer Recovery* its goal is to uncover the unknown structure of Model Graphs based on the weights of a set of input models. Our algorithm works as follows: (a) Cluster into different Model Trees based on the pairwise weight distances. (b) For each cluster, i) use $\ell_{FT}$ or $\ell_{LoRA}$ to create a pairwise distance matrix $D$ for placing edges, and ii) create a binary directional matrix $K$ based on the kurtosis to determine the edge direction. (c) To recover the final Model Tree, run a minimum directed spanning tree (MDST) algorithm on the merged prior matrix $M$. The final recovered Model Graph is the union of the recovered Model Trees

according to the pairwise weight distance. Since that node belongs to a different Model Tree, it will have a larger distance than a set threshold, allowing us to classify it as the odd one. With the node isolated, we can follow the protocol of the previous Model Trees for the two remaining nodes.

**Stranger-Stranger-Stranger (S3).** Finally, here we have a Model Graph comprising 3 Model Trees (see Fig. 3). Similar to PCS, we can cluster them into different Model Trees based on their large distances, allowing us to identify the structure.

## 5.2 MoTHer Recovery: Scaling up Model Graph Recovery

We present MoTHer, our method for recovering the structure of larger Model Graphs. Let $v_1, \ldots, v_n \in \mathcal{V}$ be a set of nodes representing different models. For simplicity, assume for now that all $v_1, \ldots, v_n \in \mathcal{T}$, i.e., all nodes are from the same Model Tree, we will later relax this assumption.

Our goal is to recover the edges $\mathcal{E}$ of the Model Tree $\mathcal{T}$, this is done by placing edges between two nodes, where one was trained from the other. As seen above, recovering the structure requires a combination of the estimated weight distance and the edge direction. Let $D$ be a weight distance matrix and let $K$ be a binary directional matrix,

$$D_{ij} = \begin{cases} \ell(v_i, v_j), & \text{if } i \neq j \\ \infty, & \text{otherwise} \end{cases}, \qquad K_{ij} = \begin{cases} 1, & \text{if } k(v_i) < k(v_j) \\ 0, & \text{otherwise} \end{cases} \qquad (4)$$

To allow for both generalization and specialization node relations, we define $T$ as a binary matrix

$$T_{ij} = \begin{cases} 1, & \text{if generalization} \\ 0, & \text{otherwise} \end{cases} \qquad (5)$$

The final distance matrix for recovering the Model Tree takes into account all 3 constraints as follows,

$$M_{ij} = D + \lambda(K \oplus T) \qquad (6)$$

where $\oplus$ is a binary XOR and $\lambda$ regularizes the directional score, allowing for some mistakes. Since $\ell_{FT}$ and $\ell_{LoRA}$ may range in value, we define $\lambda$ to be proportional to $D$ with $\lambda = c \cdot \left(\frac{1}{n^2} \sum_{i,j=1}^{n} D_{ij}\right)$ where $c$ is some constant. In practice, we found that the results are virtually unchanged for values of $c \in (0, 5)$, and chose $c = 0.3$ arbitrarily.

We can recover the Model Tree from $M$ using a minimum directed spanning tree (MDST) algorithm. In this paper, we employ the Chu-Liu-Edmonds' algorithm (Chu, 1965; Edmonds et al., 1967), which

iteratively contracts cycles in the graph until forming a tree. The algorithm proceeds as follows: initially, it treats each node as a temporary tree. Then, it merges the temporary trees via the incoming edge with the minimum weight. Subsequently, it identifies cycles in the remaining temporary trees and removes the edge with the highest weight. This merging process continues until all cycles are eliminated, resulting in the minimum directed spanning tree. The algorithm runs in $O(EV)$, however, faster MDST algorithms exist (Gabow et al., 1986) with $O(E + V \log V)$.

**Multiple components.** Thus far we assumed all $v_1, \ldots, v_n$ are from the same Model Tree. If we were to simply run the above algorithm on models that are unrelated (i.e., belong to different Model Trees), we would end up with a single, wrong Model Tree. Therefore, when dealing with general model populations, we first cluster $v_1, \ldots, v_n$ into different components based on their pairwise distances (using Eq. 1 or Eq. 2). We then run the above MoTHer recovery algorithm on each of the clusters independently. Note that while in some cases clustering is trivial by using the model architecture, it is not sufficient for the general case as there are many *unrelated* foundation models that share the exact same architecture. For instance, DINO (Caron et al., 2021), MAE (He et al., 2022), and CLIP (Radford et al., 2021) all use a VIT-B(Dosovitskiy et al., 2020) architecture despite being completely unrelated.

## 6 EXPERIMENTS

### 6.1 EXPERIMENTAL SETUP

To evaluate the performance of MoTHer we construct the *MoTHer dataset*, a Model Graph with over $500$ models organized in different Model Trees. We distinguish between $4$ main disjoint sub-graphs of the dataset: i) *LoRA-V*: LoRA fine-tuning with varying ranks, ii) *LoRA-F*: LoRA fine-tuning with fixed ranks, iii) *FT*: full fine-tuning, and iv) *Mixed*: mixed LoRA and full fine-tuning.

Each category contains $105$ models in $3$ levels of hierarchy and is comprised of $5$ Model Trees rooted by different, *unrelated* pre-trained ViTs (Dosovitskiy et al., 2020) found on *Hugging Face*. The second level of each Model Tree contains $4$ models fine-tuned on randomly chosen datasets from the VTAB benchmark(Zhai et al., 2019). Each second-level model has $4$ child nodes, fine-tuned with randomly sampled VTAB datasets while ensuring they are different than their parent model. We label all of the models as specialization-trained. In addition, we also construct a deeper Model Tree (*Deep*) with $121$ models in $5$ levels of hierarchy as well as a ResNet50(He et al., 2016) Model Tree (*ResNet*) with $21$ models in $3$ levels of hierarchy. The fine-tuning exhibits some variation in hyperparameters, for more details see App. B. In addition to the MoTHer dataset, we evaluate our method on the Stable Diffusion Model Tree found on *Hugging Face*.

We use accuracy as the evaluation metric, a correct prediction is one where both the edge placement and direction are correct. In all our tests, MoTHer ran in seconds to minutes even on a CPU (see App. F for more details). For clustering the Model Graph into different Model Trees, we use hierarchical clustering over the $\ell_2$ pairwise distance and assume knowledge of the number of clusters. We use the "scipy" (Virtanen et al., 2020) implementation with the default hyperparameters.

### 6.2 MOTHER DATASET RESULTS

We now present the results of our method. As discussed in Sec. 5.2, when given a set of models, we first cluster the models into smaller subsets where the models belong to disjoint Model Trees. We then run MoTHer recovery on the set of models in each cluster. Notably, the clustering succeeded with perfect accuracy, we therefore discuss the results of independent Model Trees. In Sec. 6.4 show the relation between the size of the Model Graph and the accuracy of the clustering.

**LoRA fine-tuning.** We start by testing MoTHer on the LoRA sub-graphs of the dataset. We set $L$ from $Eq.$ 2 and $Eq.$ 3 to be all the LoRA fine-tuned layers of the model. We first study the performance of LoRA fine-tuned models with varying ranks. Meaning, that the rank of the difference between two pairs of models is likely to be different and therefore discriminative. Indeed, as can be seen in Tab. 1, we successfully reconstruct all $5$ Model Trees within the sub-graph with perfect accuracy. In contrast, when all models use the same rank, the variance between different models decreases, resulting in a reduced accuracy for one of the $5$ Model Trees.

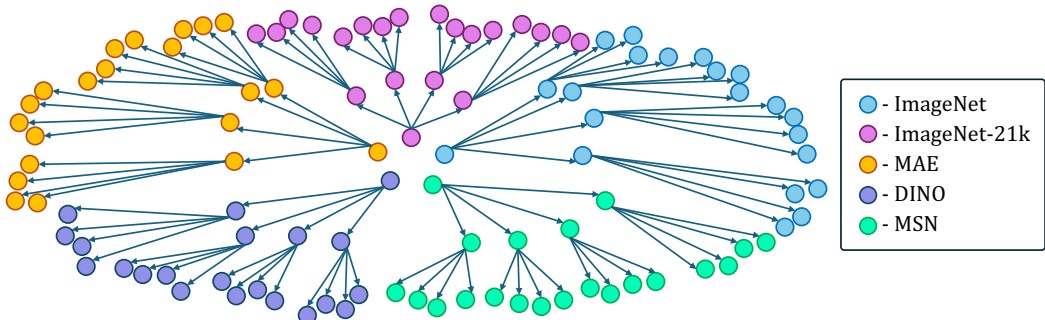

Figure 5: ***MoTHer Dataset Overview:*** Our dataset simulates a Model Graph consisting of over 20 Model Trees with a total of over 500 models fine-tuned on varying datasets with different hyperparameters. We distinguish between 4 main disjoint sub-graphs, differing in backbone and fine-tuning paradigm. We visualize the ground truth structure of a single sub-graph that contains 105 models across 5 Model Trees. The different colors represent the different Model Trees, each rooted in a different foundation model. In practice, this structure is unknown and we are only given the set of models, without knowing their relations or their origin. Note that all 105 models use the *same* ViT architecture, making it non-trivial to recover the structure

Table 1: ***MoTHer Results:*** MoTHer Recovery achieves high accuracy both for individual Model Trees and entire Model Graphs. Each sub-graph comprises 105 models from 5 Model Trees, the Mixed sub-graph simulates a real-world repository where models from the same Model Tree use either LoRA or full fine-tuning

| Sub-graph | Model Tree Root | | | | | Model |
| | ImageNet | ImageNet-21k | MAE | DINO | MSN | Graph |
| --- | --- | --- | --- | --- | --- | --- |
| LoRA-V | 1.0 | 1.0 | 1.0 | 1.0 | 1.0 | 1.0 |
| LoRA-F | 1.0 | 1.0 | 1.0 | 1.0 | 0.8 | 0.96 |
| FT | 0.85 | 0.85 | 1.0 | 1.0 | 1.0 | 0.94 |
| Mixed | 0.9 | 0.9 | 0.9 | 0.9 | 0.9 | 0.83 |

To ablate whether the LoRA-based $\ell_{LoRA}$ distance defined in Eq. 2 is necessary, we repeat the above experiment with $\ell_{FT}$ defined in $Eq.$ 1. Not using the low-rank distance prior reduces the results on *LoRA-V* by 22% to 0.78, demonstrating the significance of $\ell_{LoRA}$ for LoRA fine-tuned models.

**Full fine-tuning.** We now proceed to test our method in cases where all the models used full fine-tuning. As before, we use the already clustered sets and run MoTHer on each set independently. We set $L$ from $Eq.$ 1 to all the model layers and $Eq.$ 3 to be all the dense layers of the model. For 3 out of the 5 Model Trees in the dataset, MoTHer successfully reconstructs the tree structure with perfect accuracy. The other two Model Trees suffered from a wrong directional score, which resulted in an imperfect reconstruction, we show the full breakdown in Tab. 1. Our method also generalizes to other architectures, we demonstrate this by recovering the structure of the ResNet Model Tree with perfect accuracy.

**Mixed LoRA and full fine-tuning.** Finally, we created a sub-graph to simulate real model repositories where Model Trees contain models that use different fine-tuning paradigms. In particular, we construct a Model Graph where the fine-tuning method is randomly chosen to be either full fine-tuning or LoRA fine-tuning (with fixed rank). Since the weights of models that used full fine-tuning are full rank, we must use $Eq.$ 1 and set $L$ to be all the model layers. Similar to the drop in performance when using $\ell_{FT}$ with LoRA fine-tuned models, here too there is some decrease in performance, resulting in an overall accuracy of 0.9, see Tab. 1 for the Model Tree breakdown.

### 6.3 IN-THE-WILD MOTHER RECOVERY

Having shown we can recover the Model Graph structure with high accuracy on the MoTHer dataset, we now attempt to recover the Model Tree of in-the-wild models found on *Hugging Face*. We note

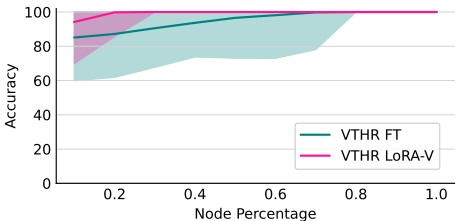

Figure 6: ***Clustering Robustness to Small Model Graphs:*** In both cases, the clustering works with high accuracy even for small Model Graphs

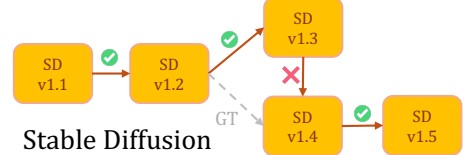

Figure 7: ***Stable Diffusion Model Tree:*** We successfully reconstruct all but a single edge, where the estimated weight distance was incorrect but the direction was correct

that the example below is special in that it provides a relatively detailed description of its model hierarchy, this provides a unique opportunity for us to test our method in a real-world situation with ground truth metrics but also emphasizes the relevance and importance of the MoTHer recovery task.

**Stable Diffusion** The *Hugging Face* model cards for Stable Diffusion describe a $4$ level hierarchy spanning $5$ of their models. These are: i) Stable Diffusion 1.1, ii) Stable Diffusion 1.2, iii) Stable Diffusion 1.3, iv) Stable Diffusion 1.4, and v) Stable Diffusion 1.5. We use the $\ell_{FT}$ distance described above, however, since the different model versions are better and more generalized foundation models we treat them as generalization nodes (i.e., the directional score is now flipped, as seed in Eq. 5). As seen in Fig. 7, MoTHer successfully reconstructs all but a single edge, incorrectly placing "Stable Diffusion 1.4" as a descendent of "Stable Diffusion 1.3", instead of a sibling. Notably, the mistake occurred due to a wrong distance, as the directional score returned the correct edge direction.

## 6.4 ABLATIONS

**Robustness to similar models.** Foundation models often come with fine-tuning "recipes", as such, many publicly available models are almost identical to each other, often with just a different seed. We therefore test the robustness of MoTHer with $3$ identically fine-tuned versions of ViT that used different seeds. In all $3$ cases, the distance between sibling models was greater than to the parent model, allowing us to correctly recover the Model Tree.

**Deeper and larger Model Trees.** We ablate whether MoTHer can scale to deeper and larger Model Trees. To this end, we train a $5$-level hierarchy of ViT models, rooted at the ImageNet foundation model. Each $i_{th}$ level model has $3$ child models, resulting in a set of $121$ models, all belonging to the same Model Tree. Indeed, although this structure has $6\times$ more nodes and is $2\times$ deeper (the root to leaf path is now $4$ edges instead of $2$), we observe a minor decrease in accuracy, from $0.85$ to $0.79$. We note that we did not test MoTHer on huge, web scale Model Graphs, in Sec. 7 we discuss scaling MoTHer recovery to web scale Model Graphs.

**Clustering robustness to smaller Model Graphs.** We now test whether the clustering succeeds for small Model Graphs. As seen in Fig. 6, even with as little as 10 models (across 5 Model Trees), the clustering achieves high accuracy, indicating MoTHer recovery could be performed even for small, yet diverse Model Graphs.

**Other directional scores.** Our directional score (Eq. 3) uses kurtosis to estimate the distributional change in outliers of weight values. However, other directional scores may be favorable. We compared the performance using the variance, skewness, kurtosis, and entropy. To do so, we fine-tuned each of the root models from the MoTHer dataset and extracted intermediate weights throughout the training process. The kurtosis is the only metric that demonstrated consistent monotonicity across the different models (see Fig. 8).

**Effect of layers types.** Neural networks often contain multiple layer types (e.g., linear, convolutional, attention). We therefore study the change in the directional score for different layer types throughout the fine-tuning process. Despite different types of layers exhibiting similar trends on average, the dense layer remained consistent across all Model Trees (see Fig. 13).

**Robustness to Pruning and Quantization.** Our method is robust to pruned (see App. D.1) and quantized models (see App. D.2). For example, with $90\%$ pruning, the accuracy decreases by only $4\%$. In the extreme case where $99\%$ of weights are pruned, our method still achieves $68\%$ accuracy

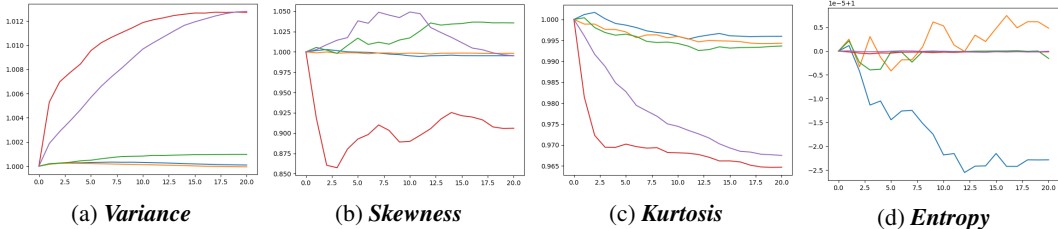

(a) *Variance*  (b) *Skewness*  (c) *Kurtosis*  (d) *Entropy*

Figure 8: *Other directional scores:* We compute the different types of directional scores throughout the fine-tuning process. The kurtosis is the only metric that remained consistently (almost) monotonic. Each color represents a fine-tuning from a different Model Tree root

(random baseline is roughly $5\%$). Moreover, when $50\%$ of the models underwent quantization, the performance of our method decreases by less than $1\%$.

## 7 DISCUSSION AND LIMITATIONS

**Training stage supervision.** Throughout the paper we assumed the training stage is known in advance. While for many use cases, this is a reasonable assumption, finding a method to automatically infer the stage based on the model weights would allow for greater applicability in cases where the training stage supervision is missing. Alternatively, one could extend MoTHer with methods for identifying the direction of an edge that does not rely on the training stage.

**MoTHer Recovery at web scale.** This paper makes the first step in recovering the Model Graph of trained neural networks. However, recovering web scale Model Graphs requires significant computational resources for storing the millions of models and computing their distance matrices. One possible solution is to train a neural network to learn compact representations that encode the relations between models. This may be done by using the limited number of already documented models as weak supervision. However, as recent studies demonstrate (Schürholt et al., 2024; Lim et al., 2023; Eilertsen et al., 2020), while the weights encode vast information about the model, treating the weights as an input to a neural network is non-trivial even for small models.

**Models with Mixed Heritage.** In this work we focused on models with a single parent. However, in some cases two or more models are merged to form a new model. In App. E we describe a preliminary experiment that explores the potential of identifying such cases where we show that it is determine forward to determine both parents of a merged model. Extending our method to fully handle such cases is left for future work.

## 8 CONCLUSION

As the number of neural network models in repositories like Hugging Face now crosses the one million threshold, tracing the heritage of a model becomes essential for attribution and can help resolve legal disputes. We analyze current model repositories and conclude that most models are poorly documented and do not contain enough information for model attribution. We propose using the Model Graph and Model Tree to organize collections of models and represent the hereditary relations between them. As the structure of the Model Graph of public repositories is unknown, we introduce the task of unsupervised MoTHer Recovery, which aims to map out unknown Model Graphs. We identify two key properties of model weights that enable the recovery of Model Graphs. We validate our approach by successfully reconstructing a Model Graph with over $500$ nodes as well as a Model Graph of in-the-wild production models. Taken together, the Model Graph and MoTHer Recovery make an exciting first step toward understanding the origin of models.

## SOCIAL IMPACT

The ability to recover model heritage trees has significant implications for intellectual property rights and ethical AI development. By providing a method to trace the heritage of fine-tuned models, this work could help resolve disputes over model ownership and identify cases of proprietary data misuse. Additionally, it promotes transparency in the AI ecosystem, potentially encouraging more responsible model sharing and development practices. However, this technique could also be misused by bad actors to reverse-engineer proprietary models or training datasets, compromising the competitive advantage or privacy safeguards of legitimate AI developers.

## REPRODUCIBILITY STATEMENT

To ensure the reproducibility of our method and results, we describe the experimental setup in Sec. 6.1. We elaborate on the exact models and hyperparameters we use in App. B. We include the our code in the supplementary material and will make it publicly available through github upon acceptance. We will also upload the entire MoTHer dataset to Hugging Face.

## ACKNOWLEDGEMENTS

This work was supported in part by the "Israel Science Foundation" (ISF), the "Council for Higher Education" (Vatat), the "Center for Interdisciplinary Data Science Research" (CIDR), the "Israeli Cyber Authority", and "KLA".

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

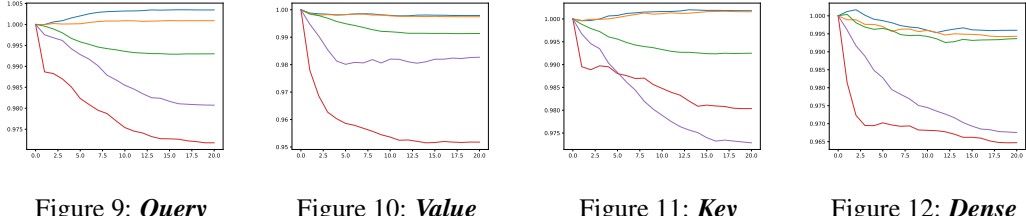

Figure 9: ***Query***   Figure 10: ***Value***   Figure 11: ***Key***   Figure 12: ***Dense***

Figure 13: ***Effect of Layer Type:*** We compute the directional score throughout the fine-tuning process. The Dense layer is the only layer that remained consistently (almost) monotonic. Each color represents a different Model Tree root

## A  MODEL CARDS ANALYSIS

To establish the benefit of using the model weights as opposed to the metadata of the model, we analyzed over $800,000$ model cards from Hugging Face. We found that at least $36\%$ of all models (roughly $290K$) do not have model cards. We used Llama 3 to analyze the remaining cards and found that for about $510K$ remaining models, about $35\%$ of model cards had no useful information about the pre-training models. *Overall, about 60% of the models (about $470K$) have no model cards or have uninformative model cards.* Even for the (about $330K$) models with "informative" cards (about $40\%$ of all models), the cards often did not describe their parent but often just the root node. As we verified the last point manually, and as there is no ground truth, we do not have an exact percentage for the models that actually have the parent node, but based on a manual inspection of $500$ randomly sampled model cards, we estimate this to be less than half of the remaining models (about $165K$). See Fig. 14 for a flowchart with the above schematic.

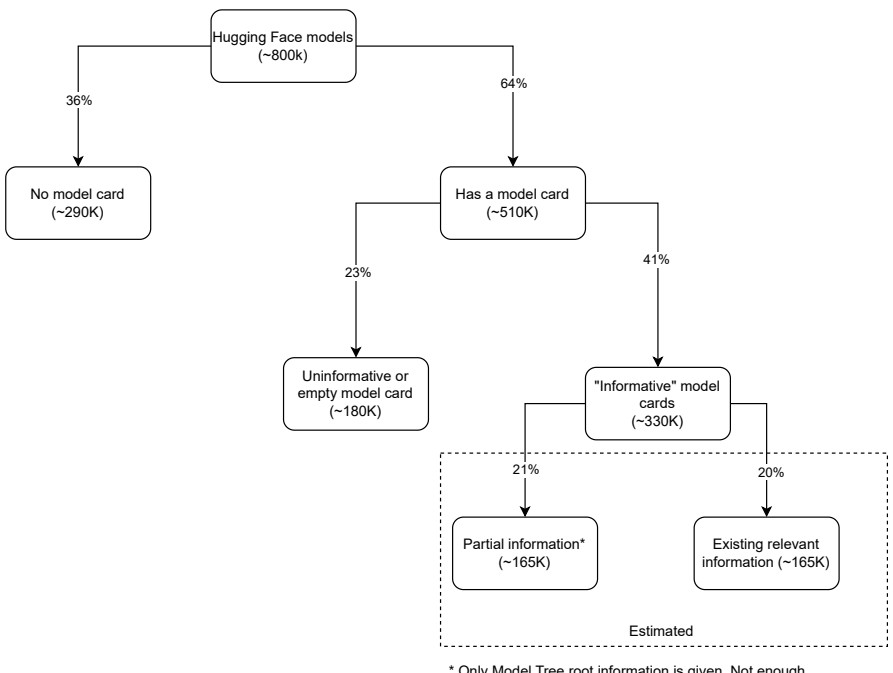

Figure 14

## B  DATASET DETAILS

For all the MoTHer dataset subsets, we use the following models as the Model Tree roots taken from *Hugging Face*:

- https://huggingface.co/google/vit-base-patch16-224
- https://huggingface.co/google/vit-base-patch16-224-in21k
- https://huggingface.co/facebook/vit-mae-base
- https://huggingface.co/facebook/dino-vitb16
- https://huggingface.co/facebook/vit-msn-base

For the FT split, to prevent model overfitting, we use larger datasets of 10K samples rather than the original 1K used in the VTAB benchmark. Each model uses a different randomly sampled seed. See Tab. 2 for additional hyperparameters.

Apart from the rank and seeds, both LoRA-F and LoRA-V use the same hyperparameters. LoRA-F uses a fixed rank and alpha of 16, LoRA-V uses ranks sampled randomly out of the options shown in Tab. 3. Both use the VTAB-1K datasets shown in Tab. 3 and random seeds. See Tab. 2 for additional hyperparameters.

Table 2: *Full Fine-tuning hyperparameters*

| Name | Value |
|---|---|
| lr | $[6e-3, 9e-3, 2e-4, 5e-4]$ |
| batch_size | 64 |
| epochs | $[2-5]$ |
| datasets | cifar100, svhn, patch_camelyon, clevr-count, clevr-distance, dmlab |

Table 3: *LoRA Varying Ranks Fine-tuning hyperparameters*

| Name | Value |
|---|---|
| lora_rank $(r)$ | $8, 16, 32, 64$ |
| lora_alpha $(\alpha)$ | $8, 16, 32, 64$ |
| lr | $[6e-3, 9e-3, 2e-4, 5e-4]$ |
| batch_size | 128 |
| epochs | $[10-20]$ |
| datasets | cifar100, caltech101, dtd, flower102, pet37, svhn, patch_camelyon, clevr-count, clevr-distance, dmlab, kitti, dsprites-location, dsprites-orientation, smallnorb-azimuth, smallnorb-elevation |

## C  STABLE DIFFUSION MODELS

For Stable Diffusion we used the following versions found on *Hugging Face*:

- https://huggingface.co/CompVis/stable-diffusion-v1-1
- https://huggingface.co/CompVis/stable-diffusion-v1-2
- https://huggingface.co/CompVis/stable-diffusion-v1-3
- https://huggingface.co/CompVis/stable-diffusion-v1-4
- https://huggingface.co/runwayml/stable-diffusion-v1-5

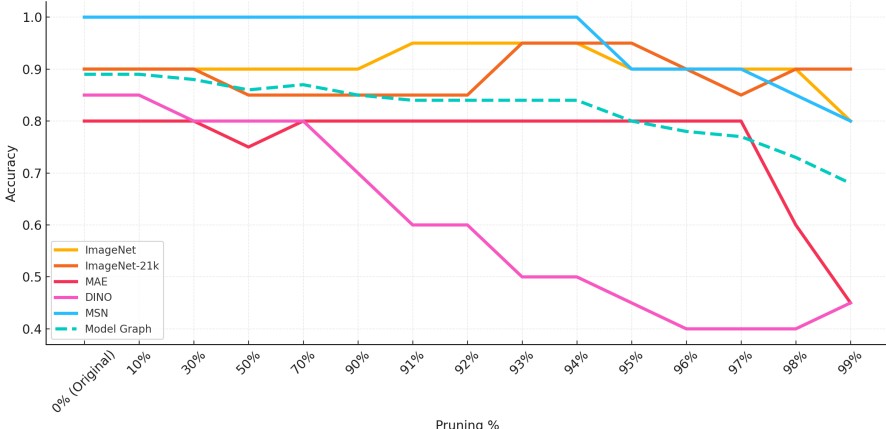

Figure 15: ***Pruning Ablation:*** Our method is robust to model pruning. For example, with $90\%$ pruning, the accuracy decreases by only $4\%$, and even at $95\%$ pruning, it drops by just $9\%$. Remarkably, when $99\%$ of weights are pruned, our method still achieves $68\%$ accuracy (random baseline is roughly $5\%$)

| Pruning % | ImageNet | ImageNet-21k | MAE | DINO | MSN | Model Graph | # Pruned Params | # Non-pruned Params |
|---|---|---|---|---|---|---|---|---|
| 0% (Original) | 0.9 | 0.9 | 0.8 | 0.85 | 1 | 0.89 | 0 | 85,524,480 |
| 10% | 0.9 | 0.9 | 0.8 | 0.85 | 1 | 0.89 | 8,552,438 | 76,972,042 |
| 30% | 0.9 | 0.9 | 0.8 | 0.8 | 1 | 0.88 | 25,657,339 | 59,867,141 |
| 50% | 0.9 | 0.85 | 0.75 | 0.8 | 1 | 0.86 | 42,762,240 | 42,762,240 |
| 70% | 0.9 | 0.85 | 0.8 | 0.8 | 1 | 0.87 | 59,867,141 | 25,657,339 |
| 90% | 0.9 | 0.85 | 0.8 | 0.7 | 1 | 0.85 | 76,972,042 | 8,552,438 |
| 91% | 0.95 | 0.85 | 0.8 | 0.6 | 1 | 0.84 | 77,827,276 | 7,697,204 |
| 92% | 0.95 | 0.85 | 0.8 | 0.6 | 1 | 0.84 | 78,682,510 | 6,841,970 |
| 93% | 0.95 | 0.95 | 0.8 | 0.5 | 1 | 0.84 | 79,537,744 | 5,986,736 |
| 94% | 0.95 | 0.95 | 0.8 | 0.5 | 1 | 0.84 | 80,393,027 | 5,131,453 |
| 95% | 0.9 | 0.95 | 0.8 | 0.45 | 0.9 | 0.8 | 81,248,261 | 4,276,219 |
| 96% | 0.9 | 0.9 | 0.8 | 0.4 | 0.9 | 0.78 | 82,103,495 | 3,420,985 |
| 97% | 0.9 | 0.85 | 0.8 | 0.4 | 0.9 | 0.77 | 82,958,729 | 2,565,751 |
| 98% | 0.9 | 0.9 | 0.6 | 0.4 | 0.85 | 0.73 | 83,814,012 | 1,710,468 |
| 99% | 0.8 | 0.9 | 0.45 | 0.45 | 0.8 | 0.68 | 84,669,246 | 855,234 |

Table 4: ***Pruning Robustness Ablation:*** Our method is robust to model pruning. For example, with $90\%$ pruning, the accuracy decreases by only $4\%$, and even at $95\%$ pruning, it drops by just $9\%$. Remarkably, when $99\%$ of weights are pruned, our method still achieves $68\%$ accuracy (random baseline is roughly $5\%$)

# D  ADDITIONAL ABLATIONS

## D.1  ROBUSTNESS TO PRUNED MODELS

We test the robustness of our method to model pruning. Specifically, we fine-tuned a new ViT model graph with a structure similar to the FT graph (5 Model Trees, each containing 21 models). We used this Model Graph and incrementally pruned weights from the models using the `l1_unstructured` function in `torch.nn.utils.prune` and evaluated our method on the pruned Model Graphs.

The results show that our method is robust to significant pruning (see Fig. 15 and a detailed breakdown in Tab. 4). For example, with $90\%$ pruning, the accuracy decreases by only $4\%$, and even at $95\%$ pruning, it drops by just $9\%$. Remarkably, when $99\%$ of weights are pruned, our method still achieves $68\%$ accuracy (random baseline is roughly $5\%$).

| Quantization Method | ImageNet | ImageNet-21k | MAE | DINO | MSN | Model Graph |
|---|---|---|---|---|---|---|
| None (Original) | 0.9 | 0.9 | 0.8 | 0.85 | 1 | 0.89 |
| fp16 | $0.9 \pm 0$ | $0.9 \pm 0$ | $0.765 \pm 0.022$ | $0.85 \pm 0$ | $1 \pm 0$ | $0.883 \pm 0.0044$ |
| Int8 | $0.9 \pm 0$ | $0.895 \pm 0.015$ | $0.77 \pm 0.024$ | $0.85 \pm 0$ | $1 \pm 0$ | $0.883 \pm 0.0078$ |

Table 5: *Quantization Robustness Ablation:* Our method is robust to weight quantization, exhibiting less than $1\%$ decrease in performance

## D.2 ROBUSTNESS TO WEIGHT QUANTIZATION

We test the robustness of our method to model quantization. Specifically, we fine-tuned a new ViT Model Graph similar to the FT graph in the paper (5 Model Trees, each containing 21 models). We then applied quantization to half of the models with a $50\%$ probability, yielding a Model Graph where $50\%$ of the models are quantized. This experiment was repeated 10 times to generate diverse quantized Model Graphs.

We tested 2 quantization methods: i) Simple quantization to fp16, ii) Int8 quantization using `bitsandbytes`. Our method is robust to weight quantization, exhibiting less than $1\%$ decrease in performance (see Tab. 5).

## D.3 DIRECTIONAL WEIGHT SCORE ANALYSIS

To further ablate the directional weight score, we fine-tuned a set of ViT models under varying learning rates and training steps. Specifically, we fine-tuned 5 models for each learning rate in the set $[1e-2, 5e-3, 1e-3, 5e-4, 1e-4, 5e-5, 1e-5, 5e-6, 1e-6]$. Each model was fine-tuned on CIFAR-100 with a unique seed for 10 epochs.

Our results show that for all models that successfully converged, the Directional Weight Score was monotonic (see Fig. 16). For the two non-convergent learning rates ($5e-3$ and $1e-2$, which achieved an accuracy below $20\%$) the Directional Weight Score no longer monotonically decreases. Notably, we observed that at some point during training (which varies across learning rates), the Directional Weight Score becomes noisy. Upon inspection, this corresponds to the model's validation loss plateauing and becoming noisy, indicating convergence.

# E MODELS WITH MIXED HERITAGE

We conducted a preliminary experiment to explore models with mixed heritage. We started with the ImageNet, MAE, and DINO pre-trained base models and merged each pair of models using standard uniform weight averaging as described in Model Soups (Wortsman et al., 2022). Subsequently, we fine-tuned 5 models from each of the original and merged models, yielding a total of 30 fine-tuned models.

We evaluate the ability of our method to handle merged models in 2 settings:

1. *Clustering:* We first clustered the 30 fine-tuned models into 6 clusters, which resulted in *perfect clustering accuracy*.

2. *Parent Detection:* To determine the parents of each merged model, we calculated the distance between each merged model and the centers of its potential parent clusters. Specifically, we computed the mean weights across all 5 fine-tuned models for each parent group (ImageNet, MAE, and DINO). For each merged fine-tuned model, we calculated the cosine similarity to each parent cluster's center.

In Tab. 6 we summarize the averaged cosine similarity between the merged models and the pre-trained clusters. As shown, the cosine similarity to the true parents is significantly higher than to the unrelated pre-trained model, which enabled us to correctly identify the parents of all merged models with *perfect accuracy*.

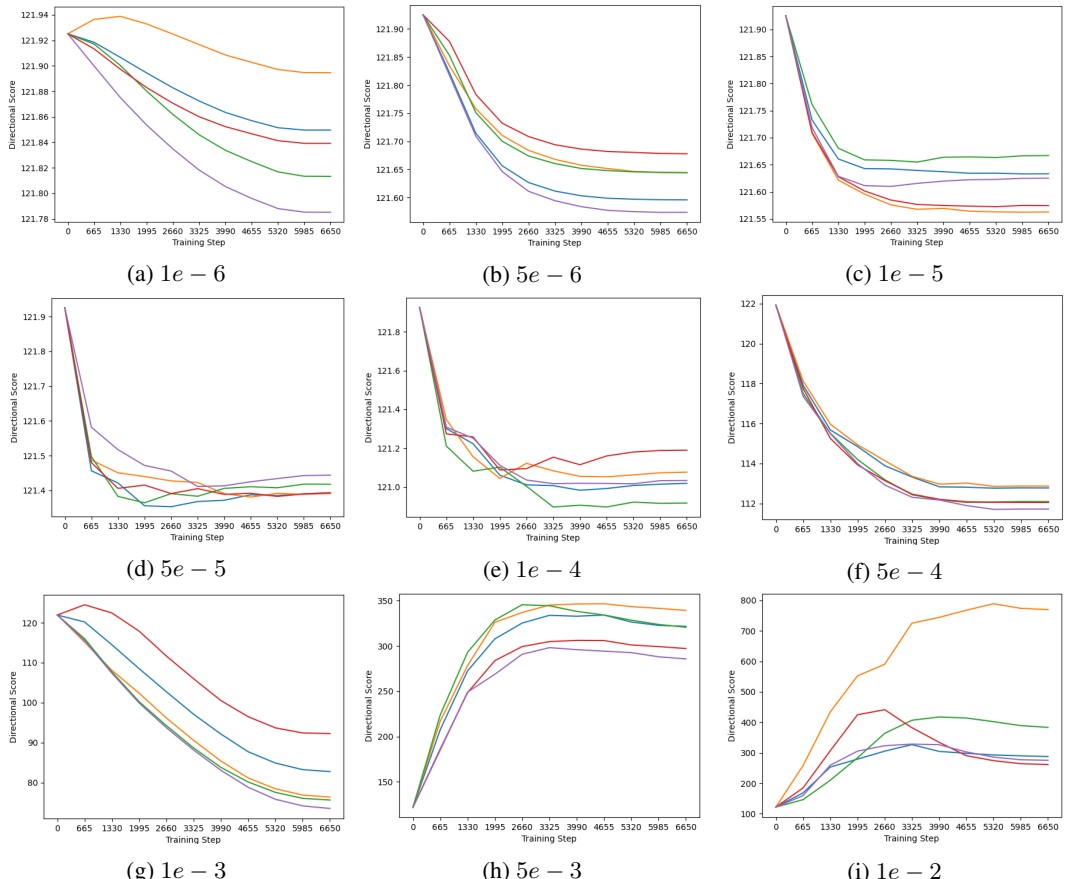

Figure 16: ***Directional Weight Score Analysis:*** We inspect the directional weight score for a wide range of learning rates and number of steps. For all models that successfully converged, the Directional Weight Score was monotonic. For the two non-convergent learning rates ($5e - 3$ and $1e-2$, which achieved an accuracy below $20\%$) the Directional Weight Score no longer monotonically decreases. Notably, we observed that at some point during training (which varies across learning rates), the Directional Weight Score becomes noisy. Upon inspection, this corresponds to the model's validation loss plateauing and becoming noisy, indicating convergence. In each sub-figure we plot 5 different models that were fine-tuned using the corresponding learning rate

|  | ImageNet | DINO | MAE |
|---|---|---|---|
| ImageNet + DINO | **0.976** | **0.192** | -0.001 |
| ImageNet + MAE | **0.847** | -0.0002 | **0.524** |
| DINO + MAE | -0.001 | **0.303** | **0.940** |

Table 6: ***Models with Mixed Heritage:*** We show the averaged cosine similarity between the merged models and the pre-trained clusters. As shown, the cosine similarity to the true parents is significantly higher than to the unrelated pre-trained model, which enabled us to correctly identify the parents of all merged models with *perfect accuracy*. In bold are the chosen parent models according to the cosine similarity

|                            | 10 Samples | 100 Samples | 1k Samples | 10k Samples |
| -------------------------- | ---------- | ----------- | ---------- | ----------- |
| Pairwise distances (CPU)   | 0.033 sec  | 0.504 sec   | 5.468 sec  | 5.01 min    |
| Pairwise distances (GPU)   | 0.1 sec    | 0.697 sec   | 0.005 sec  | 34.742 sec  |
| Clustering (CPU)           | 0.011 sec  | 0.001 sec   | 0.134 sec  | 4.43 min    |

Table 7: ***Clustering Running Time:*** Our method scales even to larger Model Graphs

## F    RUNNING TIME ANALYSIS

We measure the runtime of the clustering phase. Specifically, we simulated ViT Model Graphs of varying sizes and observed the scalability of the method. Notably, our approach allows for clustering based on the weights of a single model layer without sacrificing performance, which provides significant speedups. As can be seen, our method scales even to larger Model Graphs (see Tab. 7). Note, that the runtime of running the minimum directed spanning tree search is negligible compared to the pairwise distance calculation and clustering.

## G    THEORETICAL INTUITION FOR CLUSTERING

While the primary focus of this work is empirical, recent theoretical results provide support for the clustering of weights into trees. The literature on linear mode connectivity (LMC) has shown that models trained on the same data but with different random initializations converge to linearly related weights, up to a permutation of the neurons (Ainsworth et al., 2022). This implies that such models will be significantly distant from one another in $\ell_2$-norm in weight space.

In contrast, Frankle et al. (2020) demonstrated that models fine-tuned from a shared starting point (e.g., a pre-trained foundation model) experience less neuron permutation and tend to have weights that remain close to the original model. This establishes a clear distinction: root models in our Model Trees (typically foundation models) are far apart in weight space, whereas their fine-tuned descendants remain relatively close to their parent model.

In summary, these theoretical findings predict that intra-tree distances (between models within a tree) will be smaller than inter-tree distances (between models from different trees), thereby justifying the effectiveness of clustering models into trees.

