# OpenReview forum: "Unsupervised Model Tree Heritage Recovery"
_ICLR.cc/2025/Conference — ICLR 2025 Poster_

### Official Review · Reviewer_x9Mb · 2024-10-25

**Soundness:** 3
**Presentation:** 3
**Contribution:** 3
**Rating:** 6
**Confidence:** 4

**Summary:**

In this paper, the authors introduce the task of Unsupervised Model Tree Heritage Recovery(Unsupervised MoTHer Recovery) for collections of neural networks.

**Strengths:**

The paper is well-written and introduces the history of the model tree well.

**Weaknesses:**

The paper is good as an introduction paper. However, it seems to lack novelty in the methodology part. The dense matrix construction (6) is not new.

**Questions:**

1. You seem to use the existing graph algorithm and the model is also not new. Are there any novel points in the graph algorithm parts?
2. Could you clarify the cluster method using (1) and (2)? Do you have any guarantee of this way? Why use (1) and (2) not other criteria? In addition, is the cluster method reliable in this task? Can you use other alternative ways?

---

> ### Author Response · Authors · 2024-11-19
> **Response (1/1) to Reviewer x9Mb**
>
> We thank the reviewer for their review. Below, we provide detailed responses to the reviewer’s concerns.
> ___
>
> > The paper is good as an introduction paper. However, it seems to lack novelty in the methodology part. The dense matrix construction (6) is not new. […] existing graph algorithm and the model is also not new. Are there any novel points in the graph algorithm parts?
>
> We respectfully push back on this. The primary contributions of our paper are:
>
> 1.	The **novel task** of recovering the heritage tree of neural network models.
> 2.	The **novel directional weight score**, connecting kurtosis to weight directionality.
> 3.	Formulating our **novel task** as a minimum directional spanning tree search, incorporating a **novel cost function** that allows efficient optimization using an **established** discrete algorithm.
>
>
> Our objective (6) is new and specifically tailored to our task as it combines the weight distance and our directional score. To solve our objective we use an existing minimum directed spanning tree solver, which we found adequate for our needs and did not need to develop a new solver.
>
>
>
> ___
>
> > Could you clarify the cluster method using (1) and (2)? Do you have any guarantee of this way? Why use (1) and (2) not other criteria? In addition, is the cluster method reliable in this task? Can you use other alternative ways?
>
>
> We used standard Euclidean clustering on model weights, we validated its effectiveness in Sec. 6. We did not use (2) for clustering. Other clustering methods would probably also work, given the same distance matrix.
>
> ___
>
> We believe our response addresses all the reviewer's concerns. If the reviewer has further questions or comments, we would be happy to address them during the discussion period. If we have successfully addressed the concerns, we kindly request the reviewer to consider increasing their rating.

---

> ### Author Response · Authors · 2024-11-22
> **Follow-up**
>
> We sincerely thank you again for the time and effort you dedicated to reviewing our work, and we greatly appreciate your decision to revise the score following our rebuttal.
>
>
> Since no specific outstanding concerns were mentioned, we would be happy to discuss any remaining questions or issues if there are any. Please feel free to let us know, and we will do our best to address them.
>
>
> Thank you again for your time and for reconsidering your initial rating,
>
> The Authors

---

> > ### Comment · Reviewer_x9Mb · 2024-11-24
> > **Follow up**
> >
> > Thank you for updating the results and clarifying the problem. I will update my score accordingly.

---

> > > ### Author Response · Authors · 2024-11-24
> > >
> > > Thank you for considering our rebuttal and adjusting your score. We truly appreciate your thoughtful feedback and recognition of our work.

---

### Official Review · Reviewer_eEd1 · 2024-10-29

**Soundness:** 3
**Presentation:** 3
**Contribution:** 3
**Rating:** 6
**Confidence:** 4

**Summary:**

This paper introduces MoTHer Recovery, a method to automatically trace relationships between shared neural network models by analyzing their weights. The approach uses weight distances and distributions to determine which models were derived from others, creating a tree-like structure of model relationships without requiring training data or documentation. The authors validate their method through experiments and provide a dataset for future research in model heritage recovery.

**Strengths:**

- The paper writing is good, and presentation is clear

- The paper introduces a novel and timely problem formulation (model heritage recovery) that hasn't been systematically addressed before

- It develops an unsupervised approach that doesn't require access to training data and leverages inherent neural network weights to infer relationships

- It provides empirical validation across different fine-tuning scenarios and demonstrates effectiveness on the Stable Diffusion model family

**Weaknesses:**

- The paper doesn't address how to handle models with mixed heritage (e.g., models trained on merged weights from multiple parents) or partial weight sharing

- The clustering approach might not scale well to web-scale model repositories - needs more analysis of computational requirements

- It can be interesting to understand how different learning rates or optimization strategies during fine-tuning affect the reliability of weight-based relationships. For example, will aggressive optimization or pruning obscure these signals?

**Questions:**

- What is the computational complexity of applying this method to large model repositories? Could you provide runtime analysis for different scales (e.g., 100, 1000, 10000 models)?

- How does the method handle cases where models have been fine-tuned with different learning rates or optimization strategies? Is there a threshold where the relationship becomes undetectable?

- For models with mixed heritage (e.g., merged weights from multiple parents), how does the method determine the primary relationship? Can it detect multiple parent relationships?

---

> ### Author Response · Authors · 2024-11-19
> **Response (1/2) to Reviewer eEd1**
>
> We thank the reviewer for acknowledging our "novel and timely problem formulation." Below, we address the reviewer’s concerns in detail.
> ___
>
> > How does the method handle cases where models [...] fine-tuned with different learning rates [...]? Is there a threshold where the relationship becomes undetectable?
>
> We appreciate the reviewer's suggestion, to investigate this, we fine-tuned a set of ViT models under varying learning rates and training steps. Specifically, we fine-tuned $5$ models for each learning rate in the set $[1e-2,5e-3,1e-3,5e-4,1e-4,5e-5,1e-5,5e-6,1e-6]$. Each model was fine-tuned on CIFAR-100 with a unique seed for $10$ epochs.
>
> Our results show that for all models that successfully converged, the Directional Weight Score was monotonic. For the two non-convergent learning rates ($5e-3$ and $1e-2$, which achieved an accuracy below 20%) the Directional Weight Score no longer monotonically decreases. Notably, we observed that at some point during training (which varies across learning rates), the Directional Weight Score becomes noisy. Upon inspection, this corresponds to the model’s validation loss plateauing and becoming noisy, indicating convergence.
>
> Below, we provide the results (averaged across $5$ models per learning rate). However, **it may be difficult to observe the trend from the numbers alone. We encourage the reviewer to refer to the corresponding graphs added to App. D.3, Fig. 16 (changes highlighted in yellow), which illustrate our findings more clearly**. Each column in the table indicates the number of steps, and *each row demonstrates monotonicity for a given learning rate*.
>
> |lr|0|665|1330|1995|2660|3325|3990|4655|5320|5985|6650|
> |-|-|-|-|-|-|-|-|-|-|-|-|
> |1e-06|121.925|121.917|121.904|121.889|121.875|121.863|121.853|121.845|121.839|121.836|121.836|
> |5e-06|121.925|121.841|121.743|121.689|121.661|121.646|121.636|121.631|121.629|121.628|121.627|
> |1e-05|121.925|121.726|121.644|121.622|121.614|121.611|121.613|121.612|121.611|121.612|121.612|
> |5e-05|121.925|121.501|121.436|121.41|121.404|121.398|121.394|121.399|121.4|121.406|121.408|
> |0.0001|121.925|121.288|121.194|121.081|121.053|121.031|121.016|121.024|121.038|121.045|121.047|
> |0.0005|121.925|117.73|115.578|114.376|113.449|112.774|112.471|112.37|112.287|112.3|112.298|
> |0.001|121.925|118.276|112.029|105.8|99.721|94.108|89.004|84.888|82.106|80.571|80.134|
> |0.005|121.925|204.094|268.338|303.131|319.688|325.368|323.988|323.137|318.517|315.17|312.983|
> |0.01|121.925|183.406|292.637|369.073|404.851|433.96|425.491|414.763|408.862|398.975|395.623|
>
>
> We've also conducted experiments on quantization and pruning of the weights.
>
> For quantization, we fine-tuned a new ViT Model Graph similar to the FT graph in the paper (5 Model Trees, each containing 21 models). We then applied quantization to half of the models with a 50% probability, yielding a Model Graph where 50% of the models are quantized. This experiment was repeated 10 times to generate diverse quantized Model Graphs.
>
> We tested 2 quantization methods: i) Simple quantization to fp16, ii) Int8 quantization using `bitsandbytes`.
>
> The results show that our method is robust to quantization, as summarized below:
>
> |Quantization|ImageNet|ImageNet-21k|MAE|DINO|MSN|Model Graph|
> |-|-|-|-|-|-|-|
> |Original|0.9|0.9|0.8|0.85|1|0.89|
> |fp16|0.9±0|0.9±0|0.765±0.022|0.85±0|1±0|0.883±0.004|
> |Int8|0.9±0|0.895±0.015|0.77±0.024|0.85±0|1±0|0.883±0.007|
>
> For pruning, using the same (non-quantized) Model Graph as above, we incrementally pruned weights from the models using the `l1_unstructured` function in `torch.nn.utils.prune` and evaluated our method on the pruned Model Graphs.
>
> The results show that our method is robust to significant pruning. For example, with 90% pruning, the accuracy decreases by only 4%, and even at 95% pruning, it drops by just 9%. Remarkably, when 99% of weights are pruned, our method still achieves 68% accuracy (random baseline is roughly 5%).
>
> |Pruning %|ImageNet|ImageNet-21k|MAE|DINO|MSN|Model Graph|# Pruned Params|# Non-pruned Params|
> |-|-|-|-|-|-|-|-|-|
> |0% (Original)|0.9|0.9|0.8|0.85|1|0.89|0|85,524,480|
> |10%|0.9|0.9|0.8|0.85|1|0.89|8,552,438|76,972,042|
> |30%|0.9|0.9|0.8|0.8|1|0.88|25,657,339|59,867,141|
> |50%|0.9|0.85|0.75|0.8|1|0.86|42,762,240|42,762,240|
> |70%|0.9|0.85|0.8|0.8|1|0.87|59,867,141|25,657,339|
> |90%|0.9|0.85|0.8|0.7|1|0.85|76,972,042|8,552,438|
> |91%|0.95|0.85|0.8|0.6|1|0.84|77,827,276|7,697,204|
> |92%|0.95|0.85|0.8|0.6|1|0.84|78,682,510|6,841,970|
> |93%|0.95|0.95|0.8|0.5|1|0.84|79,537,744|5,986,736|
> |94%|0.95|0.95|0.8|0.5|1|0.84|80,393,027|5,131,453|
> |95%|0.9|0.95|0.8|0.45|0.9|0.8|81,248,261|4,276,219|
> |96%|0.9|0.9|0.8|0.4|0.9|0.78|82,103,495|3,420,985|
> |97%|0.9|0.85|0.8|0.4|0.9|0.77|82,958,729|2,565,751|
> |98%|0.9|0.9|0.6|0.4|0.85|0.73|83,814,012|1,710,468|
> |99%|0.8|0.9|0.45|0.45|0.8|0.68|84,669,246|855,234|
>
> These results have been added to Sec. 6.4 as an ablation study and elaborated upon in App. D.1 and D.2 (changes highlighted in yellow).

---

> ### Author Response · Authors · 2024-11-19
> **Response (2/2) to Reviewer eEd1**
>
> > For models with mixed heritage (e.g., merged weights from multiple parents), how does the method determine the primary relationship? Can it detect multiple parent relationships?
>
>
> Handling models with mixed heritage is beyond the primary scope of our paper. However, inspired by the reviewer’s suggestion, we conducted a preliminary experiment to explore this scenario.
>
> We started with the ImageNet, MAE, and DINO pre-trained base models and merged each pair of models using standard uniform weight averaging as described in Model Soups [1]. Subsequently, we fine-tuned $5$ models from each of the original and merged models, yielding a total of $30$ fine-tuned models.
>
> We evaluate the ability of our method to handle merged models in $2$ settings:
> 1. *Clustering:* We first clustered the $30$ fine-tuned models into $6$ clusters, which resulted in ***perfect clustering accuracy***.
> 2. *Parent Detection:* To determine the parents of each merged model, we calculated the distance between each merged model and the centers of its potential parent clusters. Specifically, we computed the mean weights across all $5$ fine-tuned models for each parent group (ImageNet, MAE, and DINO). For each merged fine-tuned model, we calculated the cosine similarity to each parent cluster’s center.
>
>
> The table below summarizes the averaged cosine similarity between the merged models (rows) and the pre-trained clusters (columns). As shown, the cosine similarity to the true parents is significantly higher than to the unrelated pre-trained model, which enabled us to ***correctly identify both parents of all merged models with perfect accuracy.*** The chosen parent models according to the cosine similarity are in bold.
>
>
> |               | ImageNet | DINO   | MAE    |
> |---------------|----------|--------|--------|
> | ImageNet + DINO | **0.976**    | **0.192**  | -0.001 |
> | ImageNet + MAE  | **0.847**    | -0.0002| **0.524**  |
> | DINO + MAE      | -0.001   | **0.303**  | **0.940**  |
>
> These results have been added to Sec. 7 as a discussion point and elaborated upon in App. E (changes highlighted in yellow).
>
> [1] Wortsman, Mitchell et al. "Model soups: averaging weights of multiple fine-tuned models improves accuracy without increasing inference time", PMLR 2022
>
> ___
>
>
> > What is the computational complexity of applying this method to large model repositories? Could you provide runtime analysis for different scales (e.g., 100, 1000, 10000 models)?
>
> Based on the reviewer’s suggestion, we report the runtime of the clustering phase. Specifically, we simulated ViT Model Graphs of varying sizes and observed the scalability of the method. Notably, our approach allows for clustering based on the weights of a single model layer without sacrificing performance, which provides significant speedups. As can be seen, our method scales even to larger Model Graphs.
>
> |                          | 10 Samples      | 100 Samples    | 1k Samples     | 10k Samples    |
> |--------------------------|-----------------|----------------|----------------|----------------|
> | Pairwise Distances (CPU) | 0.033 seconds   | 0.504 seconds  | 5.468 seconds  | 5.01 minutes   |
> | Pairwise Distances (GPU) | 0.1 seconds     | 0.697 seconds  | 0.005 seconds  | 34.742 seconds |
> | Clustering (CPU)         | 0.011 seconds   | 0.001 seconds  | 0.134 seconds  | 4.43 minutes   |
>
> The runtime of running the minimum directed spanning tree search is negligible compared to the pairwise distance calculation and clustering.
>
> These results have been added to App. F (changes highlighted in yellow).

---

> ### Author Response · Authors · 2024-11-22
> **Follow-up**
>
> We thank you again for the efforts put into reviewing our work. Like you, we believe that our paper is timely and therefore worth studying.
>
> We wanted to kindly follow up to inquire if you have had the opportunity to review our response from November 19?
>
> If there are any remaining concerns or questions, we would be happy to discuss them further and do our best to address them. If our responses have satisfactorily addressed your concerns, we would greatly appreciate your reconsideration of the score.
>
>
> Thank you,
>
> The authors

---

> > ### Comment · Reviewer_eEd1 · 2024-11-24
> >
> > Thanks for the detailed response, which addresses my comments and concerns. I would like to keep my original score.

---

> > > ### Author Response · Authors · 2024-11-24
> > >
> > > Thank you for engaging with our rebuttal and for confirming that our response addressed your comments and concerns. We truly appreciate your thoughtful review and your contributions to the discussion.
> > >
> > > If any additional questions or thoughts arise, we would be happy to continue the discussion further.
> > >
> > > Thank you again for your time and consideration,
> > >
> > > The Authors

---

### Official Review · Reviewer_Xqqo · 2024-11-01

**Soundness:** 3
**Presentation:** 3
**Contribution:** 3
**Rating:** 8
**Confidence:** 2

**Summary:**

This paper investigates the problem of finding the relationship between models from the model weights.

**Strengths:**

Please see the "Questions" section.

**Weaknesses:**

Please see the "Questions" section.

**Questions:**

My review is as follows:

- I think this paper is well-written and investigates an interesting problem.

- The introduction mentions legal disputes over model authorship. Out of curiosity, are there any known examples of this kind of dispute?

- Could you please elaborate on this point? "Moreover, it can help identify models that resulted from the wrongful use of proprietary training data." It is not clear to me how the proposed method for determining model relationships could help with wrongful use of data.

- Could the method successfully find the relationship between a quantized version of a model and the full precision model? Were there quantized models in the dataset?

- The observation that the Directional Weight Score is monotonic with respect to the training steps is interesting but perhaps not concrete enough. I would expect this to strongly depend on the specific learning rate, number of training steps, and perhaps some other hyper parameters used in training. In my opinion, identifying when this observation tends to hold and when it does not would be important in order to solidify the findings of this paper.

- Some follow-up questions on the monotonicity observation: Does this observation generalize across many different model types? On what kind of models has it been verified so far?

---

> ### Author Response · Authors · 2024-11-19
> **Response (1/3) to Reviewer Xqqo**
>
> We thank the reviewer for highlighting the strengths of our paper. Below, we provide detailed responses to the reviewer’s concerns.
>
> ___
>
>
> > The observation that the Directional Weight Score is monotonic [...] is interesting but perhaps not concrete enough. I would expect this to strongly depend on the specific learning rate, number of training steps [...]. In my opinion, identifying when this observation tends to hold [...] important in order to solidify the findings of this paper.
>
> We appreciate the reviewer’s suggestion, to investigate this, we fine-tuned a set of ViT models under varying learning rates and training steps. Specifically, we fine-tuned $5$ models for each learning rate in the set $[1e-2, 5e-3, 1e-3, 5e-4, 1e-4, 5e-5, 1e-5, 5e-6, 1e-6]$. Each model was fine-tuned on CIFAR-100 with a unique seed for $10$ epochs.
>
> Our results show that for all models that successfully converged, the Directional Weight Score was monotonic. For the two non-convergent learning rates ($5e-3$ and $1e-2$, which achieved an accuracy below 20%) the Directional Weight Score no longer monotonically decreases. Notably, we observed that at some point during training (which varies across learning rates), the Directional Weight Score becomes noisy. Upon inspection, this corresponds to the model’s validation loss plateauing and becoming noisy, indicating convergence.
>
> Below, we provide the results (averaged across $5$ models per learning rate). However, **it may be difficult to observe the trend from the numbers alone. We encourage the reviewer to refer to the corresponding graphs added to App. D.3, Fig. 16 (changes highlighted in yellow), which illustrate our findings more clearly**. Each column in the table indicates the number of steps, and *each row demonstrates monotonicity for a given learning rate*.
>
>
> |     lr |       0 |     665 |    1330 |    1995 |    2660 |    3325 |    3990 |    4655 |    5320 |    5985 |    6650 |
> |-------:|--------:|--------:|--------:|--------:|--------:|--------:|--------:|--------:|--------:|--------:|--------:|
> | 1e-06  | 121.925 | 121.917 | 121.904 | 121.889 | 121.875 | 121.863 | 121.853 | 121.845 | 121.839 | 121.836 | 121.836 |
> | 5e-06  | 121.925 | 121.841 | 121.743 | 121.689 | 121.661 | 121.646 | 121.636 | 121.631 | 121.629 | 121.628 | 121.627 |
> | 1e-05  | 121.925 | 121.726 | 121.644 | 121.622 | 121.614 | 121.611 | 121.613 | 121.612 | 121.611 | 121.612 | 121.612 |
> | 5e-05  | 121.925 | 121.501 | 121.436 | 121.41  | 121.404 | 121.398 | 121.394 | 121.399 | 121.4   | 121.406 | 121.408 |
> | 0.0001 | 121.925 | 121.288 | 121.194 | 121.081 | 121.053 | 121.031 | 121.016 | 121.024 | 121.038 | 121.045 | 121.047 |
> | 0.0005 | 121.925 | 117.73  | 115.578 | 114.376 | 113.449 | 112.774 | 112.471 | 112.37  | 112.287 | 112.3   | 112.298 |
> | 0.001  | 121.925 | 118.276 | 112.029 | 105.8   |  99.721 |  94.108 |  89.004 |  84.888 |  82.106 |  80.571 |  80.134 |
> | 0.005  | 121.925 | 204.094 | 268.338 | 303.131 | 319.688 | 325.368 | 323.988 | 323.137 | 318.517 | 315.17  | 312.983 |
> | 0.01   | 121.925 | 183.406 | 292.637 | 369.073 | 404.851 | 433.96  | 425.491 | 414.763 | 408.862 | 398.975 | 395.623 |

---

> ### Author Response · Authors · 2024-11-19
> **Response (2/3) to Reviewer Xqqo**
>
> > Could the method successfully find the relationship between a quantized version of a model and the full precision model? Were there quantized models in the dataset?
>
> The original dataset did not include quantized models. Based on the reviewer’s suggestion, we conducted additional experiments to evaluate the method on quantized models.
>
> We fine-tuned a new ViT Model Graph similar to the FT graph in the paper ($5$ Model Trees, each containing $21$ models). We then applied quantization to half of the models with a 50% probability, yielding a Model Graph where 50\% of the models are quantized. This experiment was repeated $10$ times to generate diverse quantized Model Graphs.
>
> We tested $2$ quantization methods: i) Simple quantization to fp16, ii) Int8 quantization using `bitsandbytes`.
>
> The results show that our method is robust to quantization, as summarized below:
>
>
> | Quantization Method | ImageNet         | ImageNet-21k       | MAE            | DINO           | MSN            | Model Graph    |
> |--------------|------------------|--------------------|----------------|----------------|----------------|----------------|
> | Original     | 0.9              | 0.9                | 0.8            | 0.85           | 1              | 0.89           |
> | fp16         | 0.9 ± 0          | 0.9 ± 0            | 0.765 ± 0.022  | 0.85 ± 0       | 1 ± 0          | 0.883 ± 0.0044 |
> | Int8         | 0.9 ± 0          | 0.895 ± 0.015      | 0.77 ± 0.024   | 0.85 ± 0       | 1 ± 0          | 0.883 ± 0.0078 |
>
>
>
> We've also conducted a similar experiment (requested by other reviewers) on pruned models. Specifically, we fine-tuned a new ViT model graph with a structure similar to the FT graph in the paper ($5$ Model Trees, each containing $21$ models). We used this Model Graph and incrementally pruned weights from the models using the `l1_unstructured` function in `torch.nn.utils.prune` and evaluated our method on the pruned Model Graphs.
>
> The results show that our method is robust to significant pruning. For example, with 90% pruning, the accuracy decreases by only 4%, and even at 95% pruning, it drops by just 9%. Remarkably, when 99% of weights are pruned, our method still achieves 68% accuracy (random baseline is roughly 5%).
>
>
> | Pruning %        | ImageNet | ImageNet-21k | MAE  | DINO | MSN  | Model Graph | # Pruned Params | # Non-pruned Params |
> |------------------|----------|--------------|------|------|------|-------------|-----------------|---------------------|
> | 0% (Original)    | 0.9      | 0.9          | 0.8  | 0.85 | 1    | 0.89        | 0               | 85,524,480          |
> | 10%              | 0.9      | 0.9          | 0.8  | 0.85 | 1    | 0.89        | 8,552,438       | 76,972,042          |
> | 30%              | 0.9      | 0.9          | 0.8  | 0.8  | 1    | 0.88        | 25,657,339      | 59,867,141          |
> | 50%              | 0.9      | 0.85         | 0.75 | 0.8  | 1    | 0.86        | 42,762,240      | 42,762,240          |
> | 70%              | 0.9      | 0.85         | 0.8  | 0.8  | 1    | 0.87        | 59,867,141      | 25,657,339          |
> | 90%              | 0.9      | 0.85         | 0.8  | 0.7  | 1    | 0.85        | 76,972,042      | 8,552,438           |
> | 91%              | 0.95     | 0.85         | 0.8  | 0.6  | 1    | 0.84        | 77,827,276      | 7,697,204           |
> | 92%              | 0.95     | 0.85         | 0.8  | 0.6  | 1    | 0.84        | 78,682,510      | 6,841,970           |
> | 93%              | 0.95     | 0.95         | 0.8  | 0.5  | 1    | 0.84        | 79,537,744      | 5,986,736           |
> | 94%              | 0.95     | 0.95         | 0.8  | 0.5  | 1    | 0.84        | 80,393,027      | 5,131,453           |
> | 95%              | 0.9      | 0.95         | 0.8  | 0.45 | 0.9  | 0.8         | 81,248,261      | 4,276,219           |
> | 96%              | 0.9      | 0.9          | 0.8  | 0.4  | 0.9  | 0.78        | 82,103,495      | 3,420,985           |
> | 97%              | 0.9      | 0.85         | 0.8  | 0.4  | 0.9  | 0.77        | 82,958,729      | 2,565,751           |
> | 98%              | 0.9      | 0.9          | 0.6  | 0.4  | 0.85 | 0.73        | 83,814,012      | 1,710,468           |
> | 99%              | 0.8      | 0.9          | 0.45 | 0.45 | 0.8  | 0.68        | 84,669,246      | 855,234             |
>
> These results have been added to Sec. 6.4 as an ablation study and elaborated upon in App. D.1 and D.2 (changes highlighted in yellow).

---

> ### Author Response · Authors · 2024-11-19
> **Response (3/3) to Reviewer Xqqo**
>
> > Some follow-up questions on the monotonicity observation: Does this observation generalize across many different model types? On what kind of models has it been verified so far?
>
> Yes, in the paper, we provide results for Vision Transformers (ViT), ResNet-50, and Stable Diffusion. We also tested both full fine-tuning and LoRA fine-tuning, demonstrating the robustness of the observation across diverse architectures and methods.
>
> ___
> > Could you please elaborate on this point? "Moreover, it can help identify models that resulted from the wrongful use of proprietary training data." It is not clear to me how the proposed method for determining model relationships could help with wrongful use of data.
>
> Recovering the Model Tree implicitly reveals that any data used to train an ancestor model is also part of the training data for its descendants. This has significant implications for legal disputes. For instance, if a foundation model was fine-tuned from a dataset that was later deemed to have been used improperly (e.g., lawsuits against Stable Diffusion for training on private images), any model fine-tuned from that foundation model may also violate the original data restrictions. By recovering the Model Tree, we can identify these models and trace their lineage back to the original, improperly used data.
>
>
> ___
> > The introduction mentions legal disputes over model authorship. Out of curiosity, are there any known examples of this kind of dispute?
>
> The wide spread of model sharing is quite a new phenomenon, e.g., the big jump in the number of models hosted on Hugging Face only happened over the past year. While we are not currently aware of specific legal disputes concerning model authorship, the aggressive licensing terms adopted by many model developers suggest that such litigation is likely to happen soon. For instance, the Stable Diffusion 3.5 community license restricts usage to commercial enterprises with revenues under $1 million.

---

> ### Author Response · Authors · 2024-11-22
> **Follow-up**
>
> We thank you again for the efforts put into reviewing our work. Like you, we believe that the task and our monotonicity observation are interesting.
>
> We wanted to kindly follow up to inquire if you have had the opportunity to review our response from November 19?
>
> If there are any remaining concerns or questions, we would be happy to discuss them further and do our best to address them. If our responses have satisfactorily addressed your concerns, we would greatly appreciate your reconsideration of the score.
>
>
> Thank you,
>
> The authors

---

> > ### Comment · Reviewer_Xqqo · 2024-11-25
> >
> > Thanks for the additional experiments on the quantized models and the monotonicity. My view of this paper is positive. I'll update my score.

---

> > > ### Author Response · Authors · 2024-11-26
> > >
> > > Thank you for considering our rebuttal and increasing your score. We truly appreciate your thoughtful feedback and recognition of our work.

---

### Official Review · Reviewer_VKPa · 2024-11-02

**Soundness:** 3
**Presentation:** 3
**Contribution:** 3
**Rating:** 8
**Confidence:** 3

**Summary:**

This paper targets to analyze the relation between models, aiming to shed light on which is fine-tuned from which model. This has also applications concerning copyright issues or more general licence concerns. Ths author introduces a method coined "Model Tree Heritage Recovery", which unravels the "parent-child" relations in a set of models. This method is unsupervised. Numerical examples are provided.

**Strengths:**

* Shedding light on the relation of models, in particular, in the LLM regime is crucial.
* The numerics are convincing.

**Weaknesses:**

* Due to the importance of such a method for legal aspects, some theoretical underpinning should be given, which is currently missing.
* The running time of the method is not provided.

**Questions:**

see the weaknesses

---

> ### Author Response · Authors · 2024-11-19
> **Response (1/1) to Reviewer VKPa**
>
> We thank the reviewer for acknowledging that the task can be “crucial” and for noting that the “numerics are convincing.” Below, we address the reviewer’s concerns in detail.
>
> ___
>
> > Due to the importance of such a method for legal aspects, some theoretical underpinning should be given, which is currently missing.
>
> While the primary focus of this work is empirical, recent theoretical results provide support for the clustering of weights into trees. The literature on linear mode connectivity (LMC) has shown that models trained on the same data but with different random initializations converge to linearly related weights, up to a permutation of the neurons [1]. This implies that such models will be significantly distant from one another in $\ell_2$-norm in weight space.
>
> In contrast, [2] demonstrated that models fine-tuned from a shared starting point (e.g., a pre-trained foundation model) experience less neuron permutation and tend to have weights that remain close to the original model. This establishes a clear distinction: root models in our Model Trees (typically foundation models) are far apart in weight space, whereas their fine-tuned descendants remain relatively close to their parent model.
>
> In summary, these theoretical findings predict that intra-tree distances (between models within a tree) will be smaller than inter-tree distances (between models from different trees), thereby justifying the effectiveness of clustering models into trees.
>
> We added this discussion to App. G (changes highlighted in yellow).
>
> [1] Ainsworth, Samuel K., Jonathan Hayase, and Siddhartha Srinivasa. "Git re-basin: Merging models modulo permutation symmetries." ICLR 2023.
>
> [2] Frankle, Jonathan, et al. "Linear mode connectivity and the lottery ticket hypothesis." ICML 2020.
>
>
>
>
> ___
>
> > The running time of the method is not provided.
>
> In the original manuscript, we mention that our experiments took seconds to minutes even on a CPU. Furthermore, we state that the running time for recovering the structure from a given distance matrix is  $O(EV)$. To provide a more detailed analysis, we have now conducted additional experiments to measure the runtime of the clustering phase. Specifically, we simulated ViT Model Graphs of varying sizes and observed the scalability of the method. Notably, our approach allows for clustering based on the weights of a single model layer without sacrificing performance, which provides significant speedups. As can be seen, our method scales even to larger Model Graphs.
>
> |                          | 10 Samples      | 100 Samples    | 1k Samples     | 10k Samples    |
> |--------------------------|-----------------|----------------|----------------|----------------|
> | Pairwise Distances (CPU) | 0.033 seconds   | 0.504 seconds  | 5.468 seconds  | 5.01 minutes   |
> | Pairwise Distances (GPU) | 0.1 seconds     | 0.697 seconds  | 0.005 seconds  | 34.742 seconds |
> | Clustering (CPU)         | 0.011 seconds   | 0.001 seconds  | 0.134 seconds  | 4.43 minutes   |
>
>
> The runtime of running the minimum directed spanning tree search is negligible compared to the pairwise distance calculation and clustering.
>
> These results have been added to App. F (changes highlighted in yellow).

---

> ### Author Response · Authors · 2024-11-22
> **Follow-up**
>
> We sincerely thank you again for the effort you dedicated to reviewing our work.  Like you, we believe that shedding light on the relation of models is very important.
>
> We wanted to kindly follow up to inquire if you have had the opportunity to review our response submitted on November 19.
>
> If there are any remaining concerns or questions, we would be happy to discuss them further and address them to the best of our ability. If our responses have satisfactorily addressed your concerns, we would greatly appreciate your reconsideration of the score.
>
>
> Thank you,
>
> The authors

---

> > ### Comment · Reviewer_VKPa · 2024-11-23
> > **Comment on Rebuttal**
> >
> > Dear authors, thank you for your detailed comments, which address my concerns. I will update my score accordingly.

---

> > > ### Author Response · Authors · 2024-11-24
> > >
> > > Thank you for considering our rebuttal and adjusting your score. We truly appreciate your thoughtful feedback and recognition of our work.

---

### Official Review · Reviewer_LoWV · 2024-11-04

**Soundness:** 2
**Presentation:** 2
**Contribution:** 2
**Rating:** 5
**Confidence:** 3

**Summary:**

Motivated by the fact that many models have been publicly released, this paper proposes a new problem: studying the relationships between these models. Specifically, the authors aim to build a tree data structure where directed edges connect a parent model to other models that have been directly fine-tuned from it (its children). For each pair of models, this task requires: (i) determining if they are directly related, and (ii) establishing the direction of the relationship. Assuming that all models within the model tree share the same architecture, the authors propose a method based on the distance between model weights. Experiments demonstrate the performance of the proposed method.

**Strengths:**

Originality: This paper addresses a new problem: estimating the relationship between models and their fine-tuned versions. However, the significance of this problem for open models is debatable; see the weakness for the detailed comments.

Simple approach: The proposed approach based on the distance of model weights is simple. But this is based on a well-known fact that fine-tuning makes small weight changes.

Writing: The clarity is mixed; some parts are easy to follow, but certain important sections, such as Section 4.2, are hard to understand.

**Weaknesses:**

Limitation 1: The proposed approach can only handle open models, as it relies on model weights. For important open models that have been fine-tuned, information about the pretrained models is often available at the time of release. For models without such information, one can infer relationships based on weight distance. However, it is unclear why this information is needed for all released models.

Limitation 2: The proposed approach constructs the model tree based on the weight distances between each pair of models and is thus limited to the case that all the models within a model tree share the same architecture. It can not be applied to other models that are obtained through distillation, etc.

**Questions:**

What is $\mu$ in eq. (3)? What is the pretraining stage in Figure 2 (I thought it is all about fine-tuning)? Overall, I found section 4.2 is hard to comprehend.

---

> ### Author Response · Authors · 2024-11-19
> **Response (1/2) to Reviewer LoWV**
>
> We thank the reviewer for highlighting the strengths of our paper. Below, we provide detailed responses to the reviewer’s concerns.
>
> ___
>
> > Limitation 1: The proposed approach can only handle open models, as it relies on model weights. For important open models that have been fine-tuned, information about the pretrained models is often available at the time of release. For models without such information, one can infer relationships based on weight distance. However, it is unclear why this information is needed for all released models.
>
> We break down this concern into multiple parts and address each one separately:
>
> **[...] can only handle open models [...]** While our approach relies on model weights, it is essential to differentiate between potential users of the method. Most users will indeed only have access to open models and can use our method to find fine-tuned versions of the foundation model of interest on model repositories such as Hugging Face. However, the issue of model attribution is particularly relevant in legal disputes. For instance, courts can order companies suspected of misusing a model to apply our method to their private weights, even if those weights remain undisclosed publicly.
> Consider, for example, the Stable Diffusion 3.5 community license, which restricts usage to commercial enterprises with revenue under $1 million. If Stability AI suspects a company of fine-tuning their model in violation of this license, our method could help trace the model’s origins. In such cases, courts could mandate the company to run attribution algorithms to verify compliance while the weights remain private.
>
> In summary, while the task relies on model weights, it has applications beyond open models, particularly in legal scenarios.
>
> **[...] information about the pretrained models is often available at the time of release [...]** Unfortunately, this is frequently not the case. As discussed in lines 40–45 and Appendix A, we analyzed over 800k model cards from Hugging Face and found that over 60% lack this information. This gap underscores the practical importance of our method.
>
> **[...] For models without such information, one can infer relationships based on weight distance [...]** As the reviewer noted in their summary, our task is much more complex than merely using weight distance. Our approach not only determines whether an edge exists between two models but also infers its direction. In addition to defining this task, our paper introduces a key insight: kurtosis plays a key role in determining edge directionality. We use this insight to develop our final method, MoTHer.

---

> ### Author Response · Authors · 2024-11-19
> **Response (2/2) to Reviewer LoWV**
>
> > Limitation 2: The proposed approach [...] is thus limited to the case that all the models within a model tree share the same architecture. It can not be applied to other models that are obtained through pruning, distillation, etc.
>
> The claimed limitation is not entirely accurate. While it is true that our method cannot handle distilled models (as they do not retain the original weights), it can handle pruned models.
>
> Based on the reviewer’s suggestion, we conducted an additional experiment on pruned models. Specifically, we fine-tuned a new ViT model graph with a structure similar to the FT graph in the paper (5 Model Trees, each containing 21 models). We used this Model Graph and incrementally pruned weights from the models using the `l1_unstructured` function in `torch.nn.utils.prune` and evaluated our method on the pruned Model Graphs.
>
> The results show that our method is robust to significant pruning. For example, with 90% pruning, the accuracy decreases by only 4%, and even at 95% pruning, it drops by just 9%. Remarkably, when 99% of weights are pruned, our method still achieves 68% accuracy (random baseline is roughly 5%).
>
>
> | Pruning %        | ImageNet | ImageNet-21k | MAE  | DINO | MSN  | Model Graph | # Pruned Params | # Non-pruned Params |
> |------------------|----------|--------------|------|------|------|-------------|-----------------|---------------------|
> | 0% (Original)    | 0.9      | 0.9          | 0.8  | 0.85 | 1    | 0.89        | 0               | 85,524,480          |
> | 10%              | 0.9      | 0.9          | 0.8  | 0.85 | 1    | 0.89        | 8,552,438       | 76,972,042          |
> | 30%              | 0.9      | 0.9          | 0.8  | 0.8  | 1    | 0.88        | 25,657,339      | 59,867,141          |
> | 50%              | 0.9      | 0.85         | 0.75 | 0.8  | 1    | 0.86        | 42,762,240      | 42,762,240          |
> | 70%              | 0.9      | 0.85         | 0.8  | 0.8  | 1    | 0.87        | 59,867,141      | 25,657,339          |
> | 90%              | 0.9      | 0.85         | 0.8  | 0.7  | 1    | 0.85        | 76,972,042      | 8,552,438           |
> | 91%              | 0.95     | 0.85         | 0.8  | 0.6  | 1    | 0.84        | 77,827,276      | 7,697,204           |
> | 92%              | 0.95     | 0.85         | 0.8  | 0.6  | 1    | 0.84        | 78,682,510      | 6,841,970           |
> | 93%              | 0.95     | 0.95         | 0.8  | 0.5  | 1    | 0.84        | 79,537,744      | 5,986,736           |
> | 94%              | 0.95     | 0.95         | 0.8  | 0.5  | 1    | 0.84        | 80,393,027      | 5,131,453           |
> | 95%              | 0.9      | 0.95         | 0.8  | 0.45 | 0.9  | 0.8         | 81,248,261      | 4,276,219           |
> | 96%              | 0.9      | 0.9          | 0.8  | 0.4  | 0.9  | 0.78        | 82,103,495      | 3,420,985           |
> | 97%              | 0.9      | 0.85         | 0.8  | 0.4  | 0.9  | 0.77        | 82,958,729      | 2,565,751           |
> | 98%              | 0.9      | 0.9          | 0.6  | 0.4  | 0.85 | 0.73        | 83,814,012      | 1,710,468           |
> | 99%              | 0.8      | 0.9          | 0.45 | 0.45 | 0.8  | 0.68        | 84,669,246      | 855,234             |
>
> These results have been added to Sec. 6.4 as an ablation study and elaborated upon in App. D.1 (changes highlighted in yellow).
>
> ___
>
> > What is mu in eq. (3)?
>
> Eq. (3) defines the directional weight score for computing the direction of an edge between two models. This score is based on kurtosis (fourth moment). In this equation, $\mu$ represents the mean of the layer weights $l$. We have clarified this in the updated manuscript (changes highlighted in yellow).
>
> ___
>
>  > What is the pretraining stage in Figure 2 (I thought it is all about fine-tuning)? Overall, I found section 4.2 is hard to comprehend.
>
> We appreciate this feedback and have updated the manuscript. Please let us know if further refinements are needed. Fig. 2 illustrates trends in the directional weight score during different training phases. While our paper focuses on fine-tuning, the figure highlights how the score increases during pretraining and decreases during fine-tuning.
>
> ___
>
> We believe our response addresses all the reviewer's concerns. If the reviewer has further questions or comments, we would be happy to address them during the discussion period. If we have successfully addressed the concerns, we kindly request the reviewer to consider increasing their rating.

---

> ### Author Response · Authors · 2024-11-22
> **Follow-up**
>
> We thank you again for the time and effort you dedicated to reviewing our work. We wanted to kindly follow up to inquire if you have had the opportunity to review our response from November 19?
>
> If there are any remaining concerns or questions, we would be happy to discuss them further and do our best to address them. If our responses have satisfactorily addressed your concerns, we would greatly appreciate your reconsideration of the score.
>
> Thanks,
>
> The Authors

---

> > ### Comment · Reviewer_LoWV · 2024-11-25
> >
> > I thank the authors for their efforts in addressing my comments. The additional results on pruning will strengthen the paper, although pruning does not fundamentally change the problem, as one can compare the subset of the weights. I have adjusted my score but am still unsure about the problem, which I will discuss further with other reviewers and AC.

---

> > > ### Author Response · Authors · 2024-11-25
> > >
> > > Thank you for considering our rebuttal and increasing your score, we truly appreciate your feedback.
> > >
> > > If any additional questions or thoughts arise, we would be happy to continue the discussion further.
> > >
> > > Thank you again for your time and consideration,
> > >
> > > The Authors

---

### Meta-Review · Area_Chair_ovFP · 2024-12-15

**Metareview:**

The paper proposes the task of Unsupervised Model Tree Heritage Recovery to study the relationship between different models. The idea is to build a tree where children are obtained by fine-tuning the parent, and then use weight distances and distributions to determine which models were derived from others, without requiring training data or documentation. The performance of the proposed method is assessed via numerical simulations.

The reviewers appreciated the originality of the idea, the simplicity of the approach and the convincing experiments. The main weaknesses are related to the motivation (the authors mention legal disputes but their justification during the rebuttal is not fully convincing, as they could not provide a concrete example) and to the restriction to cases where model weights are publicly available. Upon weighing strengths and limitations, I am inclined to accept the paper given the novelty of the approach that could be of interest to the ICLR community.

**Additional Comments On Reviewer Discussion:**

A few issues were raised in the reviews and most of these have been addressed -- one notable exception being the concerns of reviewer LoWV about motivation (that cannot really be addressed in the short period of the discussion).

---

### Decision · Program_Chairs · 2025-01-22

Accept (Poster)